# Safety of antidepressants in a primary care cohort of adults with obesity and depression

**Richard Morriss** [1]*, **Freya Tyrer**[2], **Francesco Zaccardi**[2], **Kamlesh Khunti**[2]

**1** Institute of Mental Health, University of Nottingham, Nottingham, United Kingdom, **2** Leicester Real World Evidence Unit, Diabetes Research Centre, University of Leicester, Leicester, United Kingdom

\* richard.morriss@nottingham.ac.uk

**Data Availability Statement:** Data cannot be shared publicly because it is owned by the NIHR CPRD on behalf of the National Health Service in England and can only be accessed by researchers making a scientific case to have access to it. Data

## Abstract

### Background

Obesity, depressive disorders and antidepressant drugs are associated with increased mortality, cardiovascular disease, diabetes, fractures and falls. We explored outcomes associated with the most commonly prescribed antidepressants in overweight or obese people with depression.

### Methods and findings

We identified a cohort of overweight or obese adults ($\geq$18 years) in primary care from the UK Clinical Practice Research Datalink, linked with hospital and mortality data, between 1 January 2000 and 31 December 2016 who developed incident depression to January 2019. Cox proportional hazards models and 99% confidence intervals were used to estimate hazard ratios (HR) for mortality, cardiovascular disease, diabetes, and falls/fractures associated with exposure to selective serotonin reuptake inhibitors (SSRIs), tricyclic (TCA)/other, combination antidepressants, citalopram, fluoxetine, sertraline, amitriptyline and mirtazapine, adjusting for potential confounding variables. In 519,513 adults, 32,350 (9.2 per 1,000 years) displayed incident depression and 21,436 (66.3%) were prescribed $\geq$1 antidepressant. Compared with no antidepressants, all antidepressant classes were associated with increased relative risks of cardiovascular disorders [SSRI HR: 1.32 (1.14–1.53), TCA/Other HR: 1.26 (1.01–1.58)], and diabetes (any type) [SSRI HR: 1.28 (1.10–1.49), TCA/Other: 1.52 (1.19–1.94)]. All commonly prescribed antidepressants except citalopram were associated with increased mortality compared with no antidepressants. However, prescription $\geq$1 year of $\geq$40mg citalopram was associated with increased mortality and falls/fractures and $\geq$1 year 100mg sertraline with increased falls/fractures.

### Conclusions

In overweight/obese people with depression, antidepressants may be overall and differentially associated with increased risks of some adverse outcomes. Further research is required to exclude indication bias and residual confounding.

are available from the Independent Scientific Advisory Committee of CPRD if researchers meet the criteria for access to confidential data. The process for obtaining such access is outlined at https://www.cprd.com/research-application. Researchers should contact the ISAC Secretariat at isac@cprd.com for further details.

**Funding:** The funding for the study came from the National Institute for Health Research (https://www.nihr.ac.uk) Collaboration for Leadership in Applied Research and Health Care East Midlands. The authors gratefully acknowledge Leicester Real-World Evidence (LRWE) Unit for providing CPRD data. LRWE Unit is funded by University of Leicester, National Institute for Health Research (NIHR) Applied Research Collaboration (ARC) East Midlands and Leicester NIHR Biomedical Research Centre. The interpretation and conclusions contained in this report/article do not necessarily reflect those of the LRWE Unit. RM is also funded by the Nottingham NIHR Biomedical Research Centre and NIHR MindTech MedTech and in-Vitro Centre. KK is supported by the NIHR Leicester Lifestyle Biomedical Research Centre (BRC) and NIHR Applied Research Collaboration (ARC-EM). The funders had no role in study design, data collection and analysis, decision to publish, or preparation of the manuscript.

**Competing interests:** The authors have declared that no competing interests exist.

## Introduction

Obesity is a leading cause of death worldwide, accounting for 44% of cases of diabetes and 23% of cardiovascular disease [1,2]. The relationship between obesity and depression is bi-directional. Being overweight/obese is associated with an increased risk of depressive disorder (27% for overweight and 55% for obesity, respectively) [3], with a dose response effect seen for increasing BMI and depression [4] and for mortality [5]. Depressive disorder is also a risk for obesity, the risk varying by gender, ethnicity, severity of depression and whether they are taking antidepressants [6–8].

An area of uncertainty is whether the prescription of antidepressants in people who have both depression and obesity may be associated with increased rates of complications from these antidepressants over and above the risks of these complications in people with obesity and depression alone. For instance, antidepressants are associated with increased falls and fractures compared with no antidepressant drugs in people with depression [9] and further increased in older obese people [10]. However, both depression and selective serotonin re-uptake inhibitor (SSRI) antidepressants lower bone mineral density [11].

Antidepressants vary substantially in the degree to which they are associated with weight gain, with the greatest weight gain associated with mirtazapine and the least with paroxetine and dosulepin [12]. However, we do not know whether the use of antidepressants in people with obesity is associated with serious adverse cardiovascular events or incident diabetes mellitus [13]. Antidepressants may show important differences in absolute rates of overall mortality [14].

There is currently no guidance on the use of antidepressants in people who are overweight/obese, although there is specific guidance on the treatment of depression in the presence of physical disease [15]. There is, therefore, a gap in knowledge in relation to the prescribing of antidepressants to people who are overweight/obese who also have depressive disorders.

Using electronic health records, we aimed to explore the drug safety of antidepressants in terms of overall mortality, cardiovascular disease (CVD), diabetes (any type), and falls/fractures in relation to the prescribing of antidepressants by class and type.

## Methods

### Data sources

We used the Clinical Practice Research Datalink (CPRD GOLD), an electronic database of more than 11.3 million patients from 674 general practices in the UK, which is broadly representative of the national population in terms of age, gender and ethnicity [16]. Approximately 75% of practices in England participate in the CPRD linkage scheme, which allows person-level linkage to hospital episode statistics (HES), deaths registrations from the Office for National Statistics (ONS) and the 2015 index of multiple deprivation (IMD) used to estimate socioeconomic status. CPRD contains Read codes and product codes for diagnoses and prescriptions whereas HES and ONS mortality data use ICD-10 codes. All of the codes used in this study have been included in the supplementary material. Independent Scientific Advisory Committee (ISAC) for CPRD approved the study, approval number 18_311R.Written/oral consent was not obtained because the data was analysed anonymously from a national database.

### Study design

This was a retrospective cohort study of patients aged 18 year or over who were overweight/obese between 1 January 2000 and 31 December 2016. Overweight/obesity was defined using either Read code for overweight/obese diagnosis (S1 Table) or BMI measurement ($\geq$25kg/m$^2$).

Inclusion criteria were meeting the "up to standard" patient and practice criteria set by the CPRD, with at least 12 months registration in the practice prior to their first overweight/obesity measurement/diagnosis in the study window, data linkage availability, and incident depression (defined using Read codes; S2 Table) during the study period.

Exclusion criteria were: antimania or antipsychotic medication (S3 Table) before date of incident depression; a diagnosis (ever) of psychotic disorder; bipolar disorder or dementia (S4 Table); anticholinesterase medication (ever) (S5 Table); or antidepressant (S6 Table), antipsychotic or antimania medication before incident depression date.

Patients were excluded if they had an overweight or obese diagnosis/measurement made after death (n = 10); of if they had an existing diagnosis of depression either: before the index date, date of registration with the practice, or before the age of 18 years (n = 11,854). They were also excluded if they had ever (either before or after the index date) a diagnosis of psychotic disorders, bipolar disorder or dementia (n = 33,643). Patients were also excluded if they were prescribed an antidepressant before the index date (n = 275,562), had ever taken anticholinesterase medication (n = 3,231), or had taken antimania or antipsychotic medication before the date of depression (n = 86,039).

## Outcome measures

The safety outcomes of interest were: (i) all-cause mortality; (ii) cardiovascular disease (including cardiovascular-related mortality); (iii) diabetes (any type); and (iv) fractures/falls. Code sets for the outcome measures were determined a-priori using primary care Read codes from CPRD Gold and version 10 codes of the International Classification of Diseases (ICD-10) for hospital episodes (admitted patient care) and mortality data. We have listed all codes for the outcome measures in S7 Table.

## Exposure measures

Antidepressant type (5 most common antidepressants only) and antidepressant drug class were defined using product codes from the CPRD and are defined in S6 Table. We included exposure of antidepressants from more than one drug or class owing to some concerns over potential adverse events.

## Covariates

Covariates were decided a priori and included: age; gender; ethnicity (white, south Asian, other, not known); overweight/obesity status (overweight [clinical diagnosis or BMI 25–29.9 kg/m$^2$], obese [clinical diagnosis or BMI 30–39.9 kg/m$^2$], severely obese [clinical diagnosis or BMI 40+ kg/m$^2$]); smoking status; alcohol status; socioeconomic status (quintile of multiple deprivation; 'not known' if missing); cancer or chronic kidney disease at baseline (i.e. depression diagnosis); glucose-lowering or statin therapy at baseline; and calendar year (2000–2004, 2005–2009, 2009–2014, 2015–2019). Read codes and ICD-10 codes for relevant covariates are described in S8 Table.

## Statistical analyses

Patients were followed up for the outcomes of interest and associations were estimated using Cox proportional hazards models with age as the timescale until their date of event, date of death or when they were last known to be alive (date of transfer out of practice, last practice update or last database linkage [22 January 2019], whichever was earliest). Exposure variables and covariates included in the model were judged to be statistically significant if they met

significance at the 1% level (p<0.01) because of the large number of variables under investigation. Therefore, 99% confidence intervals (CIs) are reported throughout. However, all covariates were left in the models even if they did not meet statistical significance because they were selected a priori as being related to both the exposures and outcome.

Antidepressant type (5 most common antidepressants only) and antidepressant drug class were included in two separate models (8 models in total for each outcome measure) as a time-dependent exposure measure to allow for changes in treatment during the follow-up period and control for immortal time bias. Adults who were not exposed to antidepressants were included as the reference category (i.e. no antidepressants ['none']). Adults were considered exposed to a drug throughout periods covered by the duration of the prescriptions if there were no gaps of more than 90 days between the end of one prescription and the start of the next prescription. If there were gaps of more than 90 days, the individual was counted as exposed to the antidepressant medication for the first 90 days of the gap and then unexposed for the remaining period.

The proportional hazards assumption was tested using Schoenfeld residuals. Where the assumption was violated, we included each of the relevant covariates as time-varying covariates (tvc) in the model using the 'tvc' option in Stata to allow the association between the covariate and mortality to change (relative to the reference category) during the follow-up period [17].

For each analysis, patients were excluded if they had an outcome of interest (cardiovascular disease, diabetes, or falls/fractures) before their first date of incident depression.

Analysis was conducted with Stata v15 [18].

### Sensitivity analyses

To assess the effect of using a presumed 90-day exposure period, we also conducted a sensitivity analysis, assuming exposure to 30 and 60 days over end of last prescription respectively. The study design involved exclusion of antipsychotic and antimania medications prior to first depression diagnosis. An additional sensitivity analysis was conducted removing any prescriptions for antipsychotic or antimania medication from the analysis.

## Results

### Baseline characteristics

Between 2000 and 2016, 32,350 obese/overweight adults had incident depression and 21,436 (66.3%) were prescribed one or more antidepressants. Table 1 shows the baseline characteristics of the study population: median age was 46.2 years (range 18–106), just over half (57%) were male, and most (84%) were white. Median follow-up was 5 years (interquartile range 2.3–8.7) and total observation period 187,369 person-years (PY). The most common antidepressants to which patients were exposed during the period were citalopram (108.2 per 1000 PY), followed by fluoxetine (63.8 per 1000 PY), sertraline (45.0 per 1000 PY), amitriptyline (20.8 per 1000 PY) and mirtazapine (10.8 per 1000 PY).

### Safety of antidepressants

Fig 1 shows the four individual models for the outcomes mortality, cardiovascular disease, diabetes and fractures/falls by drug class and Fig 2 by individual drug.

**Model 1: All-cause mortality.** There were 2,717 deaths during the observation period (14.5 per 1000 PY). Only 19 (0.7%) of these deaths were attributed to suicide. After adjustment for covariates, exposure to TCA/other antidepressants alone (HR: 1.64; 99% CI: 1.38–1.94) or to a combination of TCA/other or SSRI antidepressant or both SSRI and TCA/other

**Table 1. Baseline characteristics of study population included in all-cause mortality analysis.**

| Characteristic | Number/Median (N = 32,350*) | (Percent/IQR) |
|---|---:|---|
| Age | 46.2 | (34–59) |
| Gender*: | | |
| Male | 13,917 | -43 |
| Female | 18,433 | -57 |
| Ethnicity: | | |
| White | 27,099 | -83.8 |
| South Asian | 650 | -2 |
| Black | 375 | -1.2 |
| Mixed | 178 | -0.6 |
| Other | 324 | -1 |
| Not known | 3,724 | -11.5 |
| BMI: | | |
| Overweight (diagnosis or 25–29 kg/m$^2$) | 20,524 | -63.4 |
| Obese (diagnosis or 30–39 kg/m$^2$) | 10,049 | -31.1 |
| Severely obese (diagnosis of 40+ kg/m$^2$) | 1,777 | -5.5 |
| Smoking | | |
| Yes | 13,143 | -40.6 |
| No | 14,952 | -46.2 |
| Ex-smoker | 4,117 | -12.7 |
| Not known | 138 | -0.4 |
| Drinks alcohol (n = 313,102 available) | | |
| Yes | 5,117 | -15.8 |
| No | 12,603 | -39 |
| Ex-alcohol drinker | 904 | -2.8 |
| Not known | 13,726 | -42.4 |
| Deaths | 2,717 | -8.4 |
| Comorbidities at baseline† | | |
| Chronic kidney disease (CKD) | 2,235 | -6.9 |
| Cancer | 2,957 | -9.1 |
| Baseline medication (on or before BMI date): | | |
| Glucose lowering therapies | 3,055 | -9.4 |
| Statins | 6,179 | -19.1 |
| Index of multiple deprivation (2015) | | |
| 1 (least deprived) | 6,573 | -20.3 |
| 2 | 6,683 | -20.7 |
| 3 | 6,852 | -21.2 |
| 4 | 6,368 | -19.7 |
| 5 (most deprived) | 5,781 | -17.9 |
| Not known | 93 | -0.3 |
| Years of follow up | 5 | (2.3–8.7) |
| Antidepressant class, median time exposed (years) | | |
| Selective serotonin reuptake inhibitors | 0.8 | (0.4–2.0) |
| TCA/Other antidepressants | 0.6 | (0.3–1.5) |
| None | 3.8 | (1.4–7.1) |
| SSRI combination (2+drugs) | 0.3 | (0.2–0.3) |
| TCA/Other combination (2+ drugs) | 0.3 | (0.2–0.4) |
| TCA/Other + SRRI combination (2+drugs | 0.3 | (0.2–0.5) |

(*Continued*)

**Table 1.** (Continued)

| Characteristic | Number/Median (N = 32,350*) | (Percent/IQR) |
|---|---|---|
| Type of antidepressants, median time exposed (years): | | |
| Citalopram | 0.7 | (0.3–1.7) |
| Fluoxetine | 0.6 | (0.3–1.4) |
| Sertraline | 0.6 | (0.3–1.4) |
| Escitalopram | 0.6 | (0.3–1.3) |
| Mirtazapine | 0.4 | (0.3–1.1) |
| Paroxetine | 0.5 | (0.3–1.2) |
| Dosulepin | 0.4 | (0.3–0.9) |
| Amitriptyline | 0.4 | (0.3–1.0) |
| Venlafaxine | 0.7 | (0.3–1.9) |
| Lofepramine | 0.4 | (0.2–0.8) |
| Trazodone | 0.4 | (0.2–1.2) |
| Duloxetine | 0.5 | (0.3–1.7) |
| Other | 0.5 | (0.3–1.1) |
| SSRI combination (2+drugs) | 0.3 | (0.2–0.3) |
| TCA/Other combination (2+ drugs) | 0.3 | (0.2–0.4) |
| TCA/Other + SRRI combination (2+drugs) | 0.3 | (0.2–0.5) |

IQR Interquartile range.

* 4 people with indeterminate gender excluded; 5 people excluded because their depression occurred on the same day as date of death.

† CKD/cancer diagnosis within 10 years before depression or within 5 years post-depression.

antidepressants (HR: 2.97 [1.71–5.81] for ≥2 SSRIs; 2.18 [1.11–4.26] for ≥2 TCAs/other; and 2.98 [2.22–4.01] for SSRI and TCA/other combinations) were associated with increased all-cause mortality.

For individual antidepressants, increased all-cause mortality was associated with fluoxetine (HR: 1.70; 1.39–2.09), sertraline (HR: 1.84; CI 1.44–2.35), amitriptyline (HR: 1.76; 1.36–2.27), mirtazapine (HR: 2.11; 1.59–2.79) and other SSRIs (HR: 1.67; 1.17–2.40).

The drug class model was used to report the effects of covariates, but was similar between models. Covariates associated with all-cause mortality were: female gender (HR: 0.65; 0.59–0.73); being severely obese compared with overweight (HR: 1.58; 1.26–1.97); smoking (HR: 1.33; 1.19–1.49); being in the fourth or fifth most deprived quintile areas compared with the least deprived (HR: 1.24; 1.05–1.45; and HR: 1.41; 1.20–1.66 respectively); CKD (HR: 1.13; 1.00–1.27); cancer (HR: 234.21; 125.93–435.60 at depression diagnosis, lessening by 6% each year [tvc HR: 0.94; 0.94–0.95]); glucose lowering therapies (HR: 1.60; 1.41–1.81); and more recent calendar year periods (HR: 0.68; 0.56–0.83 for 2010–2014 and HR: 0.55; 0.44–0.68 for 2015–2019).

**Model 2: CVD.** The model for CVD included 28,544 patients, after excluding 3,806 patients with a CVD event before their first incident depression date. 1,734 (10.7 per 1000 PY) individuals had a CVD event during the observation period (PY = 162,169). After adjustment, exposure to SSRI antidepressants (HR: 1.32; 1.14–1.53), TCA/other antidepressants (HR: 1.26; 1.01–1.58) and a combination of TCA/other and SSRI antidepressants (HR: 1.86; 1.23–2.82) were associated with increased risk of CVD. For individual antidepressant drug types, citalopram (HR: 1.30; 1.07–1.57), sertraline (HR = 1.44; 1.06–1.97), and amitriptyline (HR = 1.57; 1.15–2.15) were all associated with increased risk of CVD.

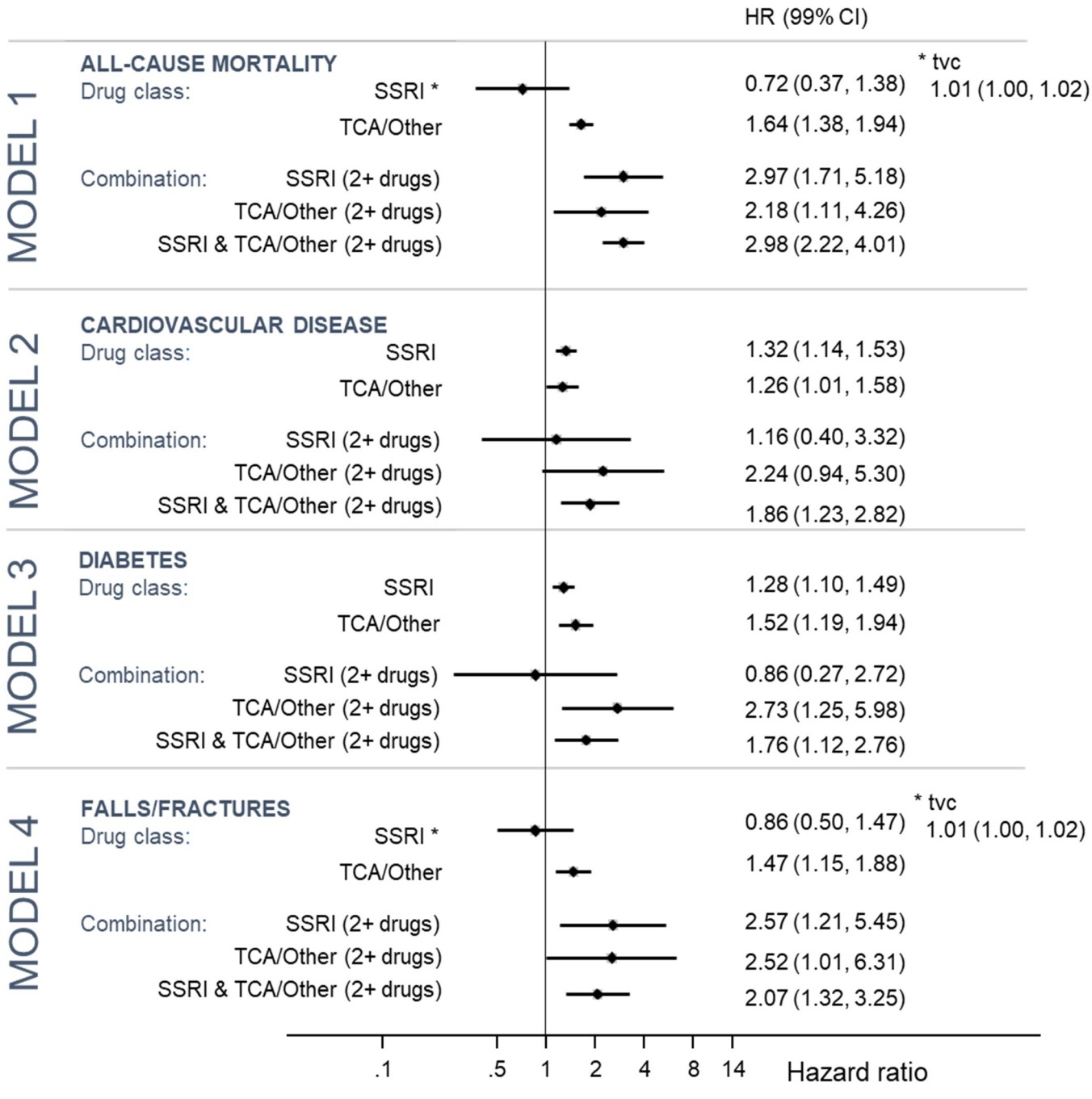

**Fig 1.**

CVD was also associated with female gender (HR: 0.22; 0.11–0.41 at depression diagnosis, increasing by 1% per year relative to male gender [tvc HR 1.01; 1.00–1.03]), being obese or severely obese compared with overweight (HR: 1.23; 1.08–1.41; and HR: 1.75; 1.37–2.24 respectively), smoking (HR: 1.24; 1.08–1.43), living in the fourth or fifth most deprived quintile areas compared with the least deprived (HR: 1.22; 1.01–1.49; and HR: 1.32; 1.08–1.61 respectively), CKD (HR: 5.87; 1.87–1.843 at diagnosis, risk decreasing by 2% each year [tvd

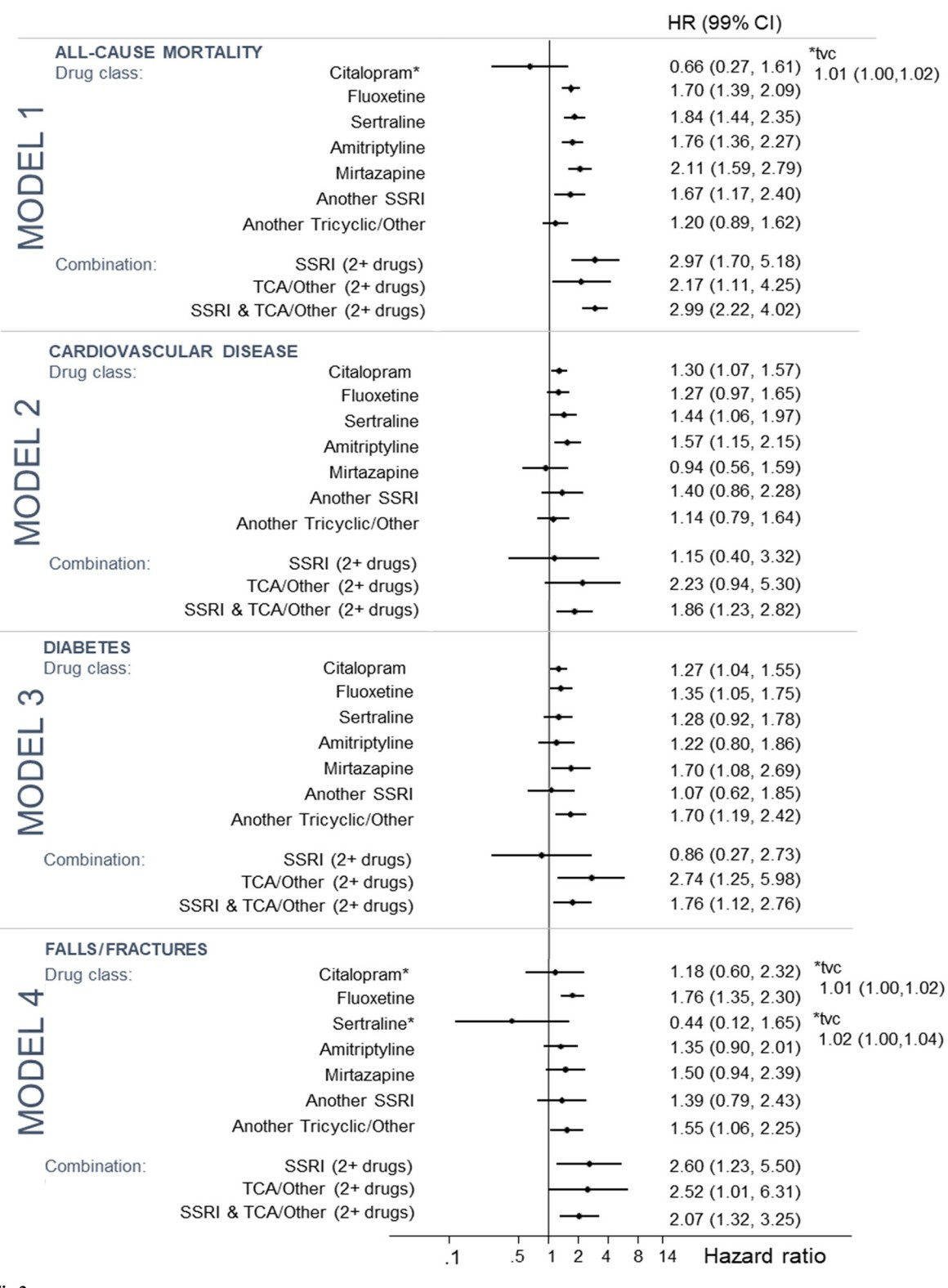

**Fig 2.**

HR: 0.97–1.00]), glucose lowering therapies (p<0.001), statin therapies (p<0.001) and more recent calendar year periods (reduced risk; p<0.001).

**Model 3: Diabetes.**   The model for diabetes included 28,152 patients, after excluding 4,198 patients with a diabetes event before their first incident depression date. 1,575 (10.0 per 1000 PY) individuals had a diabetes event during the observation period (PY = 157,691). After adjustment, exposure to SSRI antidepressants (HR: 1.28; 1.10–1.49), TCA/other antidepressants (HR: 1.52; 1.19–1.94), a combination of TCA/other antidepressants (HR: 2.73; 1.25–5.98) and a combination of TCA/other and SSRI antidepressants (HR: 1.76; 1.12–2.76) were associated with increased risk of diabetes. For individual antidepressant drug types, citalopram (p = 0.002), fluoxetine (p = 0.002), and mirtazapine (p = 0.003) were associated with increased risk of diabetes.

Covariates associated with diabetes were: female gender (HR: 0.57; 0.50–0.66; reduced risk); South Asian ethnicity compared with white (2.66; 1.81–3.91); being obese or severely obese compared with overweight (HR: 8.54; 4.42–16.48 for obese; HR: 45.89; 17.40–121.06 for severely obese at diagnosis, risk decreasing by 3% and 4% per year respectively [tvc HR: 0.98; 0.97–0.99] and 0.96; 0.94–0.98 respectively); smoking (p<0.001); living in the fifth most deprived quintile area compared with the least deprived (HR: 1.27; 1.03–1.57); CKD (HR: 1.35; 1.06–1.71); and statin therapies (HR: 1.32; 1.12–1.57).

**Model 4: Falls/Fractures.**   The model for falls/fractures included 30,959 patients, after excluding 582 patients with a fall/fracture event before their first incident depression date. A total of 1,524 (8.6 per 1000 PY) individuals had a fall/fracture during the observation period (PY = 176,559). After adjustment for the covariates, TCA/other antidepressants (HR: 1.47; 1.15–1.18), SSRI combinations (HR 2.57; 1.21–5.45); TCA/other antidepressant combinations (HR: 2.52; 1.01–6.31); and SSRI and TCA/other combinations (HR: 2.07; 1.32–3.25) were associated with increased risk of falls/fractures. Of the individual antidepressants, fluoxetine (HR: 1.76; 1.35–2.30) and other TCA/other antidepressants (HR: 1.55; 1.06–2.25) were associated with increased risk of falls/fractures.

Other covariates associated with falls/fractures were female gender (HR: 0.16; 0.10–0.27, with a time-varying effect of HR 1.02; 1.02–1.03), living in the most deprived area quintile compared with the least (HR: 1.29; 1.04–1.60) and glucose lowering therapy at baseline (HR: 1.38; 1.14–1.67).

## Effect of time-varying exposures on outcomes

The proportional hazards assumption in the models for mortality (Model 1) and fractures/falls (Model 4) were violated for both drug class and individual drug type. This effect was driven by the SSRI antidepressants citalopram (both models) and sertraline (Model 4 only). In the all-cause mortality model, the initial hazard of citalopram at baseline (i.e. first depression diagnosis) was 0.66 but increased disproportionately to the other antidepressants (in relation to the reference category no antidepressants) by 1% (HR: 1.01; 1.00–1.02) (Fig 3). Similarly, in the falls/fractures model, the initial hazard of citalopram was 1.18 and of sertraline was 0.44. Each year, these hazards increased disproportionately to the other antidepressants by 1% (HR: 1.01; 1.00–1.02) for citalopram and 2% (HR: 1.02; 1.00–1.04), respectively (Fig 3).

## Sensitivity analyses

Sensitivity analyses are summarised in Table 2. The findings were relatively robust against all sensitivity analysis scenarios with the exception of combinations of two or more TCA/other antidepressants that did not reach statistical significance at the 1% level under the 60-day exposure assumption and on removal of either antipsychotic or antimania agents. The removal of

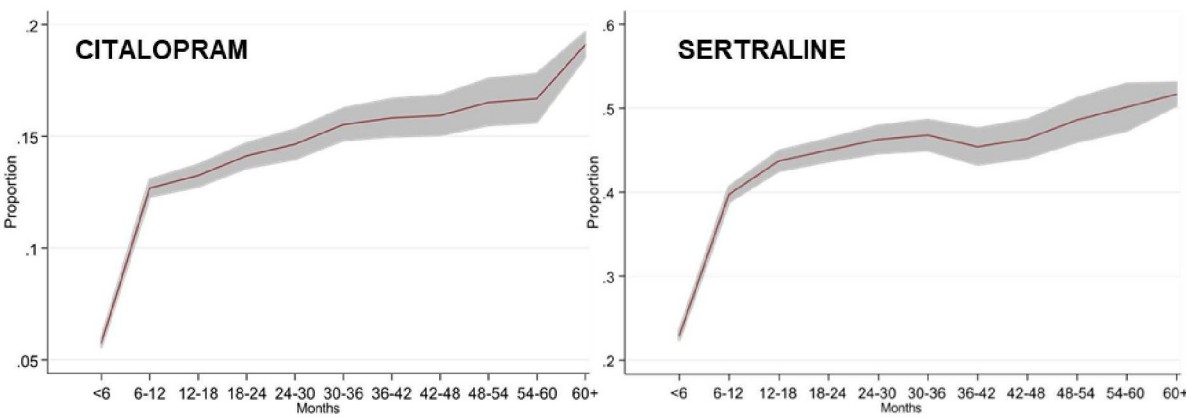

**Fig 3.**

antipsychotic agents appeared to have the greatest impact on the sensitivity analyses, such that the impact of TCA/other antidepressants (cardiovascular disease), SSRI combinations (falls/fractures), TCA/other combinations (mortality), SSRI + TCA/other combinations (falls/fractures) and mirtazapine (diabetes) no longer reached statistical significance. However, this may reflect smaller numbers in the sample (between 9.2–9.7% of patients were removed for the 4 models). We did not find an interaction between antidepressant use and overweight/obese/severely obese for any of the outcomes under investigation (p>0.01 for all).

## Discussion

Compared to a large UK primary care record study including people of any weight that was carried out over a similar time period [14], only 66% of people with depression who are overweight or obese were prescribed antidepressants compared with 88% of the general population with depression, although the duration of treatment appears similar. The five most commonly prescribed antidepressants in this database in people who were obese or overweight are the same as the general population in England, namely citalopram, fluoxetine, sertraline, amitriptyline and mirtazapine [19].

There were class effects on outcomes in obese/overweight people with incident depression, additional risks in people on combinations of antidepressants, and sometimes differing risks of antidepressants within the same class. Hence, compared with no antidepressants, SSRI or TCA/other antidepressants were associated with 26% relative increased risk of cardiovascular disease and 28% increased risk of diabetes. Exposure to combinations of SSRI and TCA/other antidepressants was associated with a significantly higher risk than single drugs for all outcomes including all-cause mortality. These findings mirror those in the general population [9,20], possibly due to indication bias in more severe and treatment resistant depression [21]. An exception was the prescription of two or more SSRIs, which were not associated with an increased risk of new cardiovascular events or diabetes. Prescribers may have been careful not to prescribe two or more SSRIs in this at risk group for cardiovascular disease and diabetes mellitus. Mirtazapine was also found to have a higher risk (70%) of diabetes than the other four antidepressants but no increased risk of cardiovascular disease compared to no antidepressants, reflecting possibly increased weight gain [12,22] but little effect on glucose or lipid homeostasis [23].

**Table 2. Summary of sensitivity analyses.**

| Exposure | Outcome | Sensitivity analyses | | | |
|---|---|---|---|---|---|
| | | Assumed exposure for 30 days after end of prescription HR (99% CI) | Assumed exposure for 60 days after end of prescription HR (99% CI) | Antipsychotic agents removed (9.2%–9.7% of total) HR (99% CI) | Antimania drugs removed (0.7% of total)HR (99% CI) |
| **Drug Class** | | | | | |
| SSRI | Mortality | - | - | - | - |
| | CVD | 1.29 (1.11–1.51) | 1.33 (1.15–1.55) | 1.35 (1.15–1.58) | 1.32 (1.14–1.53) |
| | Diabetes | 1.27 (1.08–1.48) | 1.28 (1.10–1.50) | 1.25 (1.06–1.48) | 1.28 (1.09–1.49) |
| | Falls/ fractures | - | - | - | - |
| TCA/Other | Mortality | 1.63 (1.38–1.93) | 1.63 (1.38–1.94) | 1.49 (1.22–1.82) | 1.68 (1.42–2.00) |
| | CVD | 1.30 (1.03–1.64) | 1.29 (1.03–1.62) | 1.22 (0.94–1.59) | 1.29 (1.02–1.62) |
| | Diabetes | 1.61 (1.26–2.06) | 1.56 (1.22–1.99) | 1.43 (1.08–1.90) | 1.52 (1.19–1.94) |
| | Falls/ fractures | 1.48 (1.22–1.79) | 1.47 (1.15–1.89) | 1.48 (1.12–1.96) | 1.46 (1.13–1.87) |
| SSRI combination | Mortality | 2.96 (1.54–5.66) | 2.96 (1.54–5.66) | 3.56 (1.99–6.38) | 3.07 (1.76–5.35) |
| | CVD | 0.44 (0.03–5.75) | 0.80 (0.18–3.54) | 0.72 (0.16–3.20) | 1.19 (0.41–3.41) |
| | Diabetes | 1.25 (0.28–5.57) | 1.22 (0.38–3.88) | 0.85 (0.23–3.08) | 0.87 (0.27–2.75) |
| | Falls/ fractures | 3.52 (1.67–7.42) | 3.03 (1.33–6.88) | 2.10 (0.84–5.26) | 2.41 (1.10–5.29) |
| TCA/Other combination | Mortality | 2.25 (1.12–4.51) | 2.25 (1.12–4.51) | 2.35 (0.99–5.58) | 2.42 (1.18–4.97) |
| | CVD | 1.35 (0.37–4.91) | 2.00 (0.75–5.32) | 2.29 (0.79–6.58) | 1.93 (0.73–5.13) |
| | Diabetes | 3.21 (1.41–7.29) | 2.79 (1.23–6.35) | 3.37 (1.42–8.02) | 2.70 (1.19–6.14) |
| | Falls/ fractures | 2.43 (1.08–5.42) | 2.45 (0.92–6.53) | 2.31 (0.72–7.35) | 2.42 (0.84–6.97) |
| SSRI + TCA/Other combination | Mortality | 2.78 (1.99–3.87) | 2.78 (1.99–3.87) | 3.04 (2.14–4.32) | 3.30 (2.45–4.45) |
| | CVD | 2.09 (1.30–3.38) | 1.84 (1.17–2.91) | 2.14 (1.35–3.40) | 1.96 (1.29–2.99) |
| | Diabetes | 2.14 (1.30–3.55) | 1.81 (1.11–2.94) | 1.63 (0.96–2.79) | 1.76 (1.11–2.78) |
| | Falls/ fractures | 1.99 (1.30–3.05) | 2.18 (1.35–3.51) | 1.72 (0.96–3.09) | 2.27 (1.44–3.56) |
| **Drug Type** | | | | | |
| Citalopram | Mortality | - | - | - | - |
| | CVD | 1.29 (1.06–1.57) | 1.32 (1.09–1.60) | 1.28 (1.04–1.58) | 1.30 (1.07–1.57) |
| | Diabetes | 1.25 (1.01–1.54) | 1.26 (1.03–1.55) | 1.25 (1.01–1.56) | 1.26 (1.03–1.54) |
| | Falls/ fractures | - | - | - | - |
| Fluoxetine | Mortality | 1.74 (1.42–2.14) | 1.74 (1.41–2.14) | 1.78 (1.41–2.20) | 1.76 (1.43–2.16) |
| | CVD | 1.26 (0.96–1.66) | 1.26 (0.97–1.65) | 1.39 (1.05–1.83) | 1.27 (0.97–1.65) |
| | Diabetes | 1.33 (1.02–1.75) | 1.39 (1.07–179) | 1.37 (1.04–1.80) | 1.36 (1.05–1.76) |
| | Falls/ fractures | 1.73 (1.40–2.14) | 1.75 (1.33–2.29) | 1.93 (1.46–2.55) | 1.79 (1.38–2.34) |
| Sertraline | Mortality | 1.83 (1.43–2.34) | 1.83 (1.43–2.34) | 1.83 (1.39–2.41) | 1.85 (1.45–2.36) |
| | CVD | 1.33 (0.96–1.85) | 1.45 (1.06–1.97) | 1.42 (1.01–2.00) | 1.47 (1.08–2.00) |
| | Diabetes | 1.30 (0.92–1.82) | 1.28 (0.92–1.79) | 1.19 (0.82–1.72) | 1.25 (0.90–1.75) |
| | Falls/ fractures | - | - | - | |
| Amitriptyline | Mortality | 1.77 (1.37–2.30) | 1.77 (1.37–2.30) | 1.61 (1.19–2.18) | 1.80 (1.39–2.33) |
| | CVD | 1.61 (1.15–2.24) | 1.61 (1.16–2.21) | 1.59 (1.12–2.27) | 1.57 (1.14–2.15) |
| | Diabetes | 1.35 (0.87–2.09) | 1.25 (0.81–1.92) | 1.19 (0.74–1.92) | 1.23 (0.81–1.87) |
| | Falls/ fractures | 1.25 (0.90–1.75) | 1.36 (0.90–2.04) | 1.54 (1.01–2.36) | 1.34 (0.90–2.01) |

*(Continued)*

**Table 2.** (Continued)

| Exposure | Outcome | Sensitivity analyses | | | |
|---|---|---|---|---|---|
| | | Assumed exposure for 30 days after end of prescription HR (99% CI) | Assumed exposure for 60 days after end of prescription HR (99% CI) | Antipsychotic agents removed (9.2%–9.7% of total) HR (99% CI) | Antimania drugs removed (0.7% of total) HR (99% CI) |
| Mirtazapine | Mortality | 2.08 (1.57–2.76) | 2.08 (1.57–2.76) | 1.78 (1.26–2.52) | 2.26 (1.70–3.01) |
| | CVD | 0.97 (0.57–1.63) | 0.95 (0.57–1.61) | 1.87 (0.47–1.61) | 1.01 (0.60–1.71) |
| | Diabetes | 1.80 (1.14–2.84) | 1.79 (1.14–2.81) | 1.63 (0.96–2.75) | 1.75 (1.11–2.77) |
| | Falls/ fractures | 1.54 (1.08–2.19) | 1.56 (0.98–2.47) | 1.41 (0.82–2.45) | 1.45 (0.89–2.36) |

CVD Cardiovascular disease.

The proportional hazards assumption for survival analysis was violated for SSRIs for both all-cause mortality and falls/fractures, with longer exposure to the drugs lessening the risk difference between SSRIs and other TCA/other antidepressants. Further investigation of prescriptions for citalopram and sertraline suggested that dosage increased over time (Fig 3) with a combination of higher doses and longer exposure increasing the risk of falls and fractures. Our findings in at risk groups for cardiovascular mortality and falls/fractures support the US Food and Drug Administration warning of sudden cardiac death concerning doses of citalopram ≥40mg [24] and findings in relation to lower bone density with SSRI prescription [14,25].

The main strength of the study is that we interrogated a large primary care database typical of where most of the treatment for both depression and obesity is practised. Therefore, it has considerable statistical power and generalisability to clinical practice. Randomised controlled trials of antidepressants in obese and overweight people with depression are rarely performed, are of short duration (usually 6–12 weeks) and underpowered for the detection of serious adverse effects. We included outcomes where additional serious risks might be expected in an obese population with depression. Unlike previous observational studies, we considered only incident cases of unipolar depression. The sample was representative of obese populations in terms of deprivation and ethnicity except fewer than expected people were from a South Asian population known to be at additional risk of obesity related cardiovascular disease, diabetes and death [26]. We controlled for a number of baseline confounders that might have been independently associated with the exposure and outcomes of interest including cancer and chronic kidney disease, smoking, drinking alcohol, demographic factors, deprivation and medication. We considered time, medication and length of exposure after stopping medication as sensitivity analyses.

There are important limitations to observational studies such as this. Firstly, there is confounding by indication, i.e. selective prescribing of certain types of antidepressants because of severity or chronicity of depression, associated comorbidities or symptom presentation. Confounding by indication with antidepressants may be particularly complex in relation to mortality since some antidepressants singly or in combination may be preferentially prescribed in severe and more treatment resistant depression. However in this sample and in other studies mortality from suicide was a small proportion of overall mortality in people with depression and obesity even if some suicides were not recorded as suicide but as other causes of mortality [27]. We did not have disaggregated data for other causes of mortality in this sample. It would be important to know in future research whether increased risk of diabetes and cardiovascular disease with the prescription of antidepressants for incident depression is reflected in

increased mortality from cardiovascular disease overall, by different antidepressant agent and by time or dose.

Secondly our results apply only to people who are overweight or obese and do not apply to people who are underweight or normal weight. We included people with known BMI measurements/clinical diagnoses only. Whilst complete BMI reporting in UK GP surgeries has improved over time, BMI is known to be more commonly reported in older patients, females, those with higher BMIs, people with lower socioeconomic status and people with coexisting chronic conditions [28,29]. Similarly, we have access only to primary care prescribing and not secondary care prescribing although there are no specific secondary care services for people with obesity and depression. We have no information on the severity of depression, and variation in the recognition, recording and management of patients with depression in primary care. We also did not include comparisons with the general population without overweight/ obesity and recommend this for future research.

We did not explore the adequacy of depression treatment with antidepressants, adherence to medication, and control all confounders e.g. sleep apnoea, diet, exercise, personality or other mental disorder owing to lack of available data. Some of our outcomes were aggregated and may have obscured differential effects of SSRIs and TCAs that have been found in other groups, such as on haemorrhagic and ischaemic strokes in the elderly [12,30]. Weight and BMI index are also variably recorded, especially in those who do not receive NHS Health checks [31] reducing the chances of finding real associations.

## Conclusions

When depression is recognised in obese and overweight people in primary care, GPs may be more reluctant to start antidepressant medication. There were broad class effects of antidepressants on adverse outcomes, additional risks from their combination, and some important differences among the five most commonly prescribed antidepressants, especially when prescribed at higher dosages for 12 months or more. However, some of these results need to be considered cautiously as there was likely to be indication bias and residual confounding. The results indicate the complexity of treatment of depression in people who are overweight and obese. Given the risks of both depression and continuing treatment with antidepressants for 12 months or more, especially combinations of antidepressants, a clinical review should be conducted in people who are also overweight or obese at 12 months after the last episode of depression to consider the balance of risks and benefits of continuing their current medication regime and the risks from depression. Consideration might be given to alternative effective treatments for depression such as psychological treatments should these be clinically available.

## Supporting information

**S1 Checklist. STROBE statement—Checklist of items that should be included in reports of *cohort studies*.**
(DOC)

**S1 Fig. Proportion of people on higher dose (40mg+) citalopram by length of exposure.**
(DOCX)

**S2 Fig. Proportion of people on higher dose (100mg+) sertraline by length of exposure.**
(DOCX)

**S1 Table. RECORD\* checklist.**
(DOCX)

**S2 Table. BMI measurements and Read codes for definitions of overweight, obesity and severely obese.**
(DOCX)

**S3 Table. Read codes for depression.**
(DOCX)

**S4 Table. Product codes for antipsychotic and antimania medications.**
(DOCX)

**S5 Table. Read code diagnoses of bipolar disorders, psychotic disorder or dementia.**
(DOCX)

**S6 Table. Product codes for anticholinesterase medication.**
(DOCX)

**S7 Table. Product codes for antidepressant medication.**
(DOCX)

**S8 Table. Outcome measures (Read codes and ICD-10 codes from secondary care/mortality data).**
(DOCX)

## Acknowledgments

This study is based in part on data from the Clinical Practice Research Datalink GOLD database obtained under licence from the UK Medicines and Healthcare products Regulatory Agency. However, the interpretation and conclusions contained in this report/article are those of the author/s alone and not necessarily those of the NHS, the NIHR or the Department of Health.

## Author Contributions

**Conceptualization:** Richard Morriss, Freya Tyrer, Francesco Zaccardi, Kamlesh Khunti.

**Data curation:** Freya Tyrer.

**Formal analysis:** Freya Tyrer, Francesco Zaccardi.

**Funding acquisition:** Richard Morriss, Kamlesh Khunti.

**Investigation:** Freya Tyrer.

**Methodology:** Richard Morriss, Freya Tyrer, Francesco Zaccardi, Kamlesh Khunti.

**Project administration:** Freya Tyrer.

**Resources:** Richard Morriss, Kamlesh Khunti.

**Software:** Freya Tyrer, Francesco Zaccardi.

**Supervision:** Richard Morriss, Francesco Zaccardi, Kamlesh Khunti.

**Validation:** Richard Morriss, Freya Tyrer.

**Visualization:** Richard Morriss, Freya Tyrer.

**Writing – original draft:** Richard Morriss, Kamlesh Khunti.

**Writing – review & editing:** Richard Morriss, Freya Tyrer, Francesco Zaccardi.

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
