## [Decision Letter · Decision Letter 0]

29 Apr 2020

PONE-D-20-03344

Safety of antidepressants in a primary care cohort of adults with obesity and depression

PLOS ONE

Dear Dr. Morriss,

Thank you for submitting your manuscript to PLOS ONE. After careful consideration, we feel that it has merit but does not fully meet PLOS ONE’s publication criteria as it currently stands. Therefore, we invite you to submit a revised version of the manuscript that addresses the points raised during the review process.

Of note, this paper could be improved if strategies for minimizing confounding by indication/severity as the reviewers have described have been implemented. Please also address obesity grouping and immortal time bias. The last sentence of the conclusion should be re-phrased to consider re-evaluating the continued use of antidepressant at 12 months rather than discontinue at 12 months. The Discussion section could include more description of how findings compare and add to previous studies.

We would appreciate receiving your revised manuscript by June 1, 2020. To enhance the reproducibility of your results, we recommend that if applicable you deposit your laboratory protocols in protocols.io, where a protocol can be assigned its own identifier (DOI) such that it can be cited independently in the future. For instructions see: http://journals.plos.org/plosone/s/submission-guidelines#loc-laboratory-protocols

We look forward to receiving your revised manuscript.

Kind regards,

Christine Leong, Pharm. D.

Academic Editor

PLOS ONE

Journal Requirements:

Additional Editor Comments (if provided):

Reviewers' comments:

Reviewer's Responses to Questions

**Comments to the Author**

1. Is the manuscript technically sound, and do the data support the conclusions?

Reviewer #1: Partly

Reviewer #2: Partly

2. Has the statistical analysis been performed appropriately and rigorously? 

Reviewer #1: Yes

Reviewer #2: No

3. Have the authors made all data underlying the findings in their manuscript fully available?

Reviewer #1: Yes

Reviewer #2: No

4. Is the manuscript presented in an intelligible fashion and written in standard English?

Reviewer #1: Yes

Reviewer #2: Yes

5. Review Comments to the Author

Reviewer #1: This is a study of the safety of antidepressants in a primary care cohort of adults with obesity and depression. It is set in the high quality CPRD and looks at an important and high risk population. However, despite these strengths, the study has two major areas of concern, one in its design and one in its methods, which impact the validity of its results and conclusions.

First, the design of the study, only including people who are obese and categorizing them as a single group, severely limits the authors’ ability to interpret their results and compare to the risk of mortality due to antidepressant use in the general population. The risks presented in this study are not contextualized to suggest if this population is at an elevated risk of mortality when using these medications, as is suggested in the conclusion. The empirical risk in this population is not incorrect, however, interpreting these results as a comparison relative to other populations is complicated by the lack of a direct comparison group.

- Keeping their population the same, the authors could redesign the study to understand the association between antidepressant use and the outcomes incorporating obesity as a dose-response of BMI. This would allow them to directly evaluate the interaction between antidepressant use and obesity on mortality.

Second, Model 1 (all-cause mortality) and Model 4 (suicidality) are subject to confounding by indication bias. Suicidal ideation is an indication for antidepressant use. The authors are not able to determine, with this study design, whether the risk of these outcomes affects the prescribing of antidepressants or if they are actually a consequence of antidepressant use. This leads to biased results, complicating their interpretation, and the authors mention this but are unable to estimate the magnitude of the possible error. Furthermore, if the participants in this study are at a very high risk of suicide, antidepressants could possibly lower their risk, so the residual risk remains high, but is not in fact due to their use of antidepressants, rather was underlying and a biased estimate was made apparent then interpreted as an outcome of antidepressant use. Without being able to estimate these associations, it is impossible to determine if the drugs are protective or harmful, greatly reducing the benefit of these associations to guiding care.

- The authors could consider dropping suicidality as an outcome or mentioning these results as a secondary outcome with unknown causality. Similarly, without understanding whether all cause mortality could be driven by an indication for the medication use, it is hard to interpret.

- Alternatively, they could incorporate a systematically measured index of severity of depression or suicide ideation at baseline and follow-up to contextualize if there is actually a change relative to taking antidepressants.

Additionally, in their conclusion, the authors recommend discontinuation of antidepressants after 12 months, which could undermine the effect of antidepressants if they need to be restarted. Evidence of an increased risk of suicide in the short term following initiation of antidepressants guides limiting initiation of medications as much as possible, and encouraging discontinuation could prove harmful if it leads to increased initiation. It does not appear that the authors directly evaluate this strategy (such as with dynamic treatment regimens) to improve this recommendation nor due they present a formal risk/benefit analysis for discontinuation. This remains conjectural until more careful analysis is done and should be noted as such.

Overall, this is a good paper and provides some important evidence on CVD and fracture outcomes. There remain some crucial concerns in this paper that need to be addressed to improve the validity and utility of this study, or a need to focus it more directly on the outcomes less susceptible to confounding by indication.

Reviewer #2: Major comments:

1. I would like to get a clearer picture on how the authors accounted for immortal time bias. It is not clear from the manuscript as it is written if it was appropriately considered.

2. The use of forward selection strategy to adjust for confounders is not recommended and not reasonable given the sample size. A directed acyclic graph will be a more robust approach, or PS analysis.

3.There was no attempt to minimize confounding by severity, for example adjusting for number of visits in the prior year or number of switches

4. Table 1 should be divided to include comparison group.

other:

how Weight and BMI index are also variably recorded?? this could be a major issue

The conclusion should be rewritten to account for the major limitation, indication bias

6. PLOS authors have the option to publish the peer review history of their article (what does this mean?). If published, this will include your full peer review and any attached files.

Reviewer #1: No

Reviewer #2: No

---

## [Author Response · Author response to Decision Letter 0]

5 Jun 2020

RESPONSE TO REVIEWER 1

The risks presented in this study are not contextualized to suggest if this population is at an elevated risk of mortality when using these medications, as is suggested in the conclusion. The empirical risk in this population is not incorrect, however, interpreting these results as a comparison relative to other populations is complicated by the lack of a direct comparison group.

We thank the reviewer for this comment. We have now added in a statement and corresponding references to the Introduction (first paragraph, line 50) to highlight that there is a dose-response relationship between BMI and depression, and also between BMI and mortality. We hope that this contextualises more fully that depression is a particular challenge in this population and that treatment approaches need to consider potential risks associated with depression and obesity.

In terms of comparisons with the general population, we agree that it is a limitation that we were unable to make comparisons with the general population in the ‘normal’ (and underweight) BMI category. We have now highlighted this in the Discussion (please see limitations section, lines 390-391).

- Keeping their population the same, the authors could redesign the study to understand the association between antidepressant use and the outcomes incorporating obesity as a dose-response of BMI. This would allow them to directly evaluate the interaction between antidepressant use and obesity on mortality.

We did not find an interaction between antidepressant use and overweight/obese/severely obese for any of the outcomes under investigation (p>0.01 for all) – see results line 312. We can confirm that we adjusted for the relationship between BMI (categorised as overweight, obese and severely obese) in this analysis. We have now made this clearer (see covariates sub-heading / Methods, line 152). 

Second, Model 1 (all-cause mortality) and Model 4 (suicidality) are subject to confounding by indication bias. Suicidal ideation is an indication for antidepressant use. The authors are not able to determine, with this study design, whether the risk of these outcomes affects the prescribing of antidepressants or if they are actually a consequence of antidepressant use. This leads to biased results, complicating their interpretation, and the authors mention this but are unable to estimate the magnitude of the possible error. Furthermore, if the participants in this study are at a very high risk of suicide, antidepressants could possibly lower their risk, so the residual risk remains high, but is not in fact due to their use of antidepressants, rather was underlying and a biased estimate was made apparent then interpreted as an outcome of antidepressant use. Without being able to estimate these associations, it is impossible to determine if the drugs are protective or harmful, greatly reducing the benefit of these associations to guiding care.

We have discussed this issue more thoroughly in the limitation section of the Discussion (lines 374-386). Confounding by indication with antidepressants in relation to suicidality and mortality is complex. On the one hand an obese person with depression at high suicide risk is more likely to be prescribed antidepressants. However a person who self-harms by overdose with little intent to end their life is less likely to be prescribed antidepressants in case they took the antidepressants as an overdose. Given that the risk of self-harm is 40 times greater than suicide in people who are obese (reference 28 in the paper), then there is likely to be confounding by indication but the confounding is operating to both increase and decrease the likelihood of antidepressant prescribing. 

In relation to mortality, suicide is a rare outcome compared to deaths from other causes such as cardiovascular disease and cancer. While a high risk of death by suicide would certainly increase the likelihood that antidepressants would be prescribed, a risk of mortality due to other possible causes would deter the prescription of antidepressants e.g. people with obesity who are prescribed opiate drugs, antidepressant prescribing would be discouraged because of the risk of mortality from respiratory depression.

- The authors could consider dropping suicidality as an outcome or mentioning these results as a secondary outcome with unknown causality. Similarly, without understanding whether all cause mortality could be driven by an indication for the medication use, it is hard to interpret.

We agree that that results of the analysis in relation to suicide and mortality is not straight forward because of confounding by indication that may be operating to both increase and decrease these risks. We have outlined these in the Discussion (lines 374 to 386). We describe these sources of confounding and what impact they may have in the discussion. 

- Alternatively, they could incorporate a systematically measured index of severity of depression or suicide ideation at baseline and follow-up to contextualize if there is actually a change relative to taking antidepressants.

We agree that ideally we would include a measure of severity of depression but we have not have not found a suitable proxy measure for this. The reviewer suggested number of switches of antidepressants (which would be subject to immortal time bias given that the patient has to have survived long enough in order to have a switch) or number of consultations prior to the depression diagnosis (which would interact with the adjustments for glucose-lowering therapies, statins, CKD and cancer diagnosis). We have acknowledged that inability to account for depression severity is a significant limitation of our study (lines 374-386).

Additionally, in their conclusion, the authors recommend discontinuation of antidepressants after 12 months, which could undermine the effect of antidepressants if they need to be restarted. Evidence of an increased risk of suicide in the short term following initiation of antidepressants guides limiting initiation of medications as much as possible, and encouraging discontinuation could prove harmful if it leads to increased initiation. It does not appear that the authors directly evaluate this strategy (such as with dynamic treatment regimens) to improve this recommendation nor due they present a formal risk/benefit analysis for discontinuation. This remains conjectural until more careful analysis is done and should be noted as such.

We agree that the conclusion should be changed to a recommendation to clinically review each patient after being on antidepressants for 12 months. The clinical consideration is the need to manage depression and its risks by remaining on antidepressants or start an alternative treatment for depression e.g. psychological treatment and the risk of harm from antidepressants. We have altered our conclusion accordingly (lines 409-412). 

Overall, this is a good paper and provides some important evidence on CVD and fracture outcomes. There remain some crucial concerns in this paper that need to be addressed to improve the validity and utility of this study, or a need to focus it more directly on the outcomes less susceptible to confounding by indication.

We thank the reviewer for these comments.

RESPONSE TO REVIEWER 2 

Major comments:

1. I would like to get a clearer picture on how the authors accounted for immortal time bias. It is not clear from the manuscript as it is written if it was appropriately considered.

We used a time-dependent approach (for our exposure measure, antidepressant use) to control for immortal time bias (see papers [1] and [2] below for justification of this approach). This choice means that a patient who is not exposed to an antidepressant is treated as ‘unexposed’ (i.e. no antidepressant at that time period – our reference category ‘none’). We have now clarified this more fully in the statistical analyses section [Methods, lines 152-154].

[1] Zhou et al. (2005) https://academic.oup.com/aje/article/162/10/1016/65057

[2] Suissa (2007). https://academic.oup.com/aje/article/167/4/492/233064

2. The use of forward selection strategy to adjust for confounders is not recommended and not reasonable given the sample size. A directed acyclic graph will be a more robust approach, or PS analysis.

We would like to reassure the reviewer that the confounders/covariates were selected a priori as being conceptually related to the outcome of interest and not through forward selection. We did not draw a DAG specifically, but a similar approach was adopted. A PS analysis may have been possible, but would have been complex, given the different number of antidepressants under investigation. We would have needed to define the probability of being prescribed each individual antidepressant, and censoring on switch date, thereby compromising length of follow up. We are confident that our approach is sufficiently robust.

3.There was no attempt to minimize confounding by severity, for example adjusting for number of visits in the prior year or number of switches

We acknowledge that controlling for severity of depression is a limitation of our study. However, adding in number of switches would introduce immortal time bias (given that a person has to have survived/be event-free in order to have a switch). Similarly, there will be an interaction between number of consultations and glucose-lowering therapies, cancer, CKD and statins. We are reluctant to add in additional covariates at this stage as this was neither an a priori hypothesis nor in the approved ISAC protocol. However, we have acknowledged in the discussion that severity of depression was not possible as a limitation of the study (lines 374 to 386).

4. Table 1 should be divided to include comparison group.

Given that antidepressant exposure was treated as a time-dependent measure, there is no comparison group – only those who are not exposed during the time period. For example, a patient who enters the study at age 30 (i.e. first date of depression diagnosis) and who is initially prescribed no antidepressants for 1 year followed by sertraline for 1 year before leaving the practice contributes 1 year (age 30) to the “none” category (i.e. non-exposed) and 1 year (age 31) to the sertraline category. Therefore, the same patients contribute to both the reference and exposure category. We hope that this is now clearer with the new narrative added (Statistical analysis section lines 152-154).

other:

how Weight and BMI index are also variably recorded?? this could be a major issue

We agree that this is a limitation of the study, given that weight is variably recorded on the CPRD and is not missing at random. We have now added this to the limitations section (please see Discussion, lines 382-386) and have added in two new references to support this.

Of particular note, we included only�people with known 

BMI measurement. Complete BMI reporting in UK GP surgeries, 

which has improved over time, is known to be more common in older 

individuals, females, people with lower socioeconomic status� and 

higher BMIs, and people with coexisting chronic conditions (43,44

The conclusion should be rewritten to account for the major limitation, indication bias

We agree. We have added the statement “However some of these results need to be considered cautiously as there was likely to be indication bias and residual confounding (lines406-407)” to our conclusion.

EDITORIAL TEAM

We note that you have indicated that data from this study are available upon request. PLOS only allows data to be available upon request if there are legal or ethical restrictions on sharing data publicly. For information on unacceptable data access restrictions, please see http://journals.plos.org/plosone/s/data-availability#loc-unacceptable-data-access-restrictions.

There is a legal restriction on the authors sharing a de-identified data set from CPRD. There is on-going data collection of psuedoanonymised data from primary care practices, hospitals and public health as a national resource by CPRD on behalf of the National Institute for Health Research in England. The authors of the manuscript applied for, paid and were loaned the data for a set period of time after which they do not have legal access to the data. This has been set legally for the data protection of the participants and staff who collect the data because the information that is kept is very detailed, personal and sensitive including detailed patterned activity that over time might identify participants. There are also such safeguards in place because participants have their data entered into the database without formal consent; participants are only allowed to opt out so there are additional safeguards on access to the data in line with the United Kingdom Data Protection Act and GPDR. The data operates on this basis to maximise population coverage and data completeness to minimise selection, non-responder and attrition bias, thereby enhancing the generalizability of findings to clinical practice. Access to the database requires the development of a suitable protocol that is peer reviewed and approved by a board. A key principle is that there has to be a strong scientific case for the use of the data and that only the data is released to researchers with the expertise to utilise it to answer the research. The researchers also need to provide adequate assurances around data protection and data security.

However, researchers internationally can apply for access to any data including the same data that we used. The process for obtaining such access is outlined at https://www.cprd.com/research-applications (accessed 05/28/2020).

---

## [Decision Letter · Decision Letter 1]

11 Nov 2020

PONE-D-20-03344R1

Safety of antidepressants in a primary care cohort of adults with obesity and depression

PLOS ONE

Dear Dr. Morriss,

Thank you for submitting your manuscript to PLOS ONE. After careful consideration, we feel that it has merit but does not fully meet PLOS ONE’s publication criteria as it currently stands. Therefore, we invite you to submit a revised version of the manuscript that addresses the points raised during the review process.

I would like to apologize that this took a bit longer than expected (mainly due to the COVID care that has to be provided at this moment). One of the two reviewers has addressed some methodological points that need to be addressed before considering the manuscript for publication.

We look forward to receiving your revised manuscript.

Kind regards,

Nienke van Rein

Academic Editor

PLOS ONE

Reviewers' comments:

Reviewer's Responses to Questions

**Comments to the Author**

1. If the authors have adequately addressed your comments raised in a previous round of review and you feel that this manuscript is now acceptable for publication, you may indicate that here to bypass the “Comments to the Author” section, enter your conflict of interest statement in the “Confidential to Editor” section, and submit your "Accept" recommendation.

Reviewer #1: (No Response)

Reviewer #2: All comments have been addressed

2. Is the manuscript technically sound, and do the data support the conclusions?

Reviewer #1: No

Reviewer #2: Yes

3. Has the statistical analysis been performed appropriately and rigorously? 

Reviewer #1: No

Reviewer #2: Yes

4. Have the authors made all data underlying the findings in their manuscript fully available?

Reviewer #1: No

Reviewer #2: No

5. Is the manuscript presented in an intelligible fashion and written in standard English?

Reviewer #1: Yes

Reviewer #2: Yes

6. Review Comments to the Author

Reviewer #1: This paper investigates adverse outcomes associated with anti-depressant use in a large cohort of overweight and obese adults. While there are some useful results from the study, there continue to be serious issues with the authors’ approach and conclusions.

The context of this paper, as a study among people with overweight or obese BMI, is overreached to suggest that the observed associations may exist among the general population (a group including people with underweight or normal weight BMI measures). The authors should make it much clearer that their results are observed in a population only including overweight and obese individuals. If the authors suggest that their study sample is representative, then their results are dissimilar from the literature exploring similar questions, which includes RCTs (PMID: 27367876, PMID: 32617669), and potential sources of confounding and bias should be further investigated. Even the randomized studies and meta-analyses of randomized studies that find short term risk, show much lower risk levels – which suggests that this estimate is only sensible if there is no increase in suicide among participants who are not overweight. This would be a major finding, if true, but more likely there is confounding by indication.

This paper would be stronger (and not subject to the bias it currently is) if the authors only included CVD, diabetes, and falls/fractures outcomes in their analyses/results. Suicide and all-cause mortality are subject to serious confounding by indication, which the authors agree with, and it is unclear what the value of giving a biased estimate is to the literature. Given that the study cohort is relatively young (mean age: 46, 25% under 34 years) and less likely to experience some of the outcomes related to age, premature death, such as due to suicide, should be a consideration in the methods and particular attention should be made to limit confounding in these analyses. The authors’ inability to account for severity of depression prevents them from attaining unbiased results from models that are subject to confounding related to this measure. Additionally, the inclusion of individuals taking multiple antidepressants (e.g. 2+ SSRIs) exacerbates the confounding by indication since individuals are requiring a second medication, potentially an indicator for severe depression and suicide ideation. The authors report very high hazard ratios for the risk of suicide among these individuals, which are likely severely confounded. All-cause mortality encompasses suicide, thus is subject to the same issues.

The authors slightly modified their conclusion; however, their recommendation for possible discontinuation of antidepressants is an unsubstantiated claim based off their study. An interesting study could have been designed on the benefits or risk of discontinuing medication, where there is an absence of randomized evidence. This conclusion could even prove to be harmful, as there is evidence of adverse impact on risk of suicide in the short-term following antidepressant initiation and current recommendations suggest minimizing the amount of times initiating antidepressants. The conclusion should be further adjusted to not suggest discontinuation of antidepressants.

Overall, this paper contains important content related to antidepressants and CVD, diabetes, and falls/fractures, but the authors present biased estimates that need to be addressed before this paper provides a beneficial contribution to the literature

Reviewer #2: If confounders were selected as a priori and adjusted for in full models and not using forward selection, then this should be stated and considered the superior method.

No additional comments

7. PLOS authors have the option to publish the peer review history of their article (what does this mean?). If published, this will include your full peer review and any attached files.

Reviewer #1: No

Reviewer #2: No

---

## [Author Response · Author response to Decision Letter 1]

19 Nov 2020

Response to Reviewers

Reviewer #1: This paper investigates adverse outcomes associated with anti-depressant use in a large cohort of overweight and obese adults. While there are some useful results from the study, there continue to be serious issues with the authors’ approach and conclusions.

The context of this paper, as a study among people with overweight or obese BMI, is overreached to suggest that the observed associations may exist among the general population (a group including people with underweight or normal weight BMI measures). The authors should make it much clearer that their results are observed in a population only including overweight and obese individuals. If the authors suggest that their study sample is representative, then their results are dissimilar from the literature exploring similar questions, which includes RCTs (PMID: 27367876, PMID: 32617669), and potential sources of confounding and bias should be further investigated. Even the randomized studies and meta-analyses of randomized studies that find short term risk, show much lower risk levels – which suggests that this estimate is only sensible if there is no increase in suicide among participants who are not overweight. This would be a major finding, if true, but more likely there is confounding by indication.

Reply. We have added some text to the discussion and conclusion that further explains that the findings apply to obese and overweight people by:

1. Line 373 removing the reference to the representativeness of the sample to the general population.

2. Lines 401-402. Adding a statement that the results apply only to obese and overweight people not normal or underweight people.

3. Line 430. State that the clinical review at 12 months should be conducted in obese/overweight people.

The abstract makes no reference to the general population and its conclusion only mentions people who are obese or overweight.

This paper would be stronger (and not subject to the bias it currently is) if the authors only included CVD, diabetes, and falls/fractures outcomes in their analyses/results. Suicide and all-cause mortality are subject to serious confounding by indication, which the authors agree with, and it is unclear what the value of giving a biased estimate is to the literature. Given that the study cohort is relatively young (mean age: 46, 25% under 34 years) and less likely to experience some of the outcomes related to age, premature death, such as due to suicide, should be a consideration in the methods and particular attention should be made to limit confounding in these analyses. The authors’ inability to account for severity of depression prevents them from attaining unbiased results from models that are subject to confounding related to this measure. Additionally, the inclusion of individuals taking multiple antidepressants (e.g. 2+ SSRIs) exacerbates the confounding by indication since individuals are requiring a second medication, potentially an indicator for severe depression and suicide ideation. The authors report very high hazard ratios for the risk of suicide among these individuals, which are likely severely confounded. All-cause mortality encompasses suicide, thus is subject to the same issues.

Reply. We have removed all data for suicide/self-harm from the paper because we agree that there may be serious confounding by indication so there are now four outcomes. However, we think that all cause morality is extremely unlikely to be seriously confounded by indication because only 19 (0.7%) of 2717 deaths were attributed to suicide taking all sources of information available to us from the database. Even if there were a substantial number of other deaths due to suicide that were attributed to other causes, then it is still unlikely that there would be serious confounding. Another database study cited in the paper also founds that suicide is a relatively uncommon cause of death in overweight or obese with depression. Unfortunately apart from suicide we cannot say what the specific causes of death were. Given that the SMR for cardiovascular death in depression is approximately 2, obesity is also a risk factor for cardiovascular death and trials of antidepressants have not reduced cardiovascular events, then it is plausible that many of these young deaths might be due to cardiovascular causes.

In summary we have removed references to suicide and self-harm from the abstract, introduction, aims, method, results, Figures 1 and 2, Table 2 and discussion. Under all-cause mortality results we have added the data that there were 19 suicides out of 2717 deaths (lines 212-213). We have acknowledged the possibility that there may be more deaths from suicide than are reported but have also stated that since suicide is a rare outcome in this and other database studies of overweight and obese people with depression (reference number 27), then we consider the risk of serious confounding by indication to be low (lines 386-398). We acknowledge that we do not know the cause of death other than suicide in the participants of this study and this would be important to study. 

The authors slightly modified their conclusion; however, their recommendation for possible discontinuation of antidepressants is an unsubstantiated claim based off their study. An interesting study could have been designed on the benefits or risk of discontinuing medication, where there is an absence of randomized evidence. This conclusion could even prove to be harmful, as there is evidence of adverse impact on risk of suicide in the short-term following antidepressant initiation and current recommendations suggest minimizing the amount of times initiating antidepressants. The conclusion should be further adjusted to not suggest discontinuation of antidepressants.

Reply. We have further modified our conclusion (lines 430-434) that people who are overweight or obese with depression who are taking antidepressants should be clinically reviewed to consider the balance of risks and benefits of continuing antidepressants in view of the risks from both the antidepressants and depression. We have also added that alternative effective treatments for depression such as psychological treatments should be considered if they are available. We think our data support such a conclusion and we are not advocating that on the basis of this data that everyone should stop taking antidepressants after one year. Furthermore in many countries, including the UK, these conclusions would not be controversial as they are recommended practice for people with mental disorder whatever their weight. Access to psychological treatments is readily available. However we know that this is not the case in many countries so we hope that we have now struck the correct clinical balance in our conclusion. 

Overall, this paper contains important content related to antidepressants and CVD, diabetes, and falls/fractures, but the authors present biased estimates that need to be addressed before this paper provides a beneficial contribution to the literature.

Reply. We hope the above addresses these concerns.

Reviewer #2: If confounders were selected as a priori and adjusted for in full models and not using forward selection, then this should be stated and considered the superior method.

Reply. We thank the reviewer for pointing this out. We can confirm that we adjusted for the covariates selected a priori in all models and retained the covariates even if they were not statistically significant in line with the a priori selection. We have now clarified this (Line 153–155). 

Have the authors made all data underlying the findings in their manuscript fully available? The PLOS Data policy requires authors to make all data underlying the findings described in their manuscript fully available without restriction, with rare exception (please refer to the Data Availability Statement in the manuscript PDF file). The data should be provided as part of the manuscript or its supporting information, or deposited to a public repository. For example, in addition to summary statistics, the data points behind means, medians and variance measures should be available. If there are restrictions on publicly sharing data—e.g. participant privacy or use of data from a third party—those must be specified.

Reply. We have already explained that there are restrictions on publicly sharing data because the data is owned by a third party and licensed for a period of time for a fee to complete a defined protocol of work that is peer reviewed and agreed between the guardians of this publicly owned data and the research team. Anyone who wishes to check or use the data in this study would need to apply to the guardians of the data and go through a similar process and pay an appropriate fee to obtain access to such data. We have outlined these procedures in our last revision. Unfortunately we are unable to modify these procedures or provide greater access.

---

## [Decision Letter · Decision Letter 2]

7 Jan 2021

Safety of antidepressants in a primary care cohort of adults with obesity and depression

PONE-D-20-03344R2

Dear Dr. Morriss,

We’re pleased to inform you that your manuscript has been judged scientifically suitable for publication and will be formally accepted for publication once it meets all outstanding technical requirements.

Kind regards,

Nienke van Rein

Academic Editor

PLOS ONE

Additional Editor Comments (optional):

Reviewers' comments:

Reviewer's Responses to Questions

**Comments to the Author**

1. If the authors have adequately addressed your comments raised in a previous round of review and you feel that this manuscript is now acceptable for publication, you may indicate that here to bypass the “Comments to the Author” section, enter your conflict of interest statement in the “Confidential to Editor” section, and submit your "Accept" recommendation.

Reviewer #1: All comments have been addressed

2. Is the manuscript technically sound, and do the data support the conclusions?

Reviewer #1: Yes

3. Has the statistical analysis been performed appropriately and rigorously? 

Reviewer #1: Yes

4. Have the authors made all data underlying the findings in their manuscript fully available?

Reviewer #1: No

5. Is the manuscript presented in an intelligible fashion and written in standard English?

Reviewer #1: Yes

6. Review Comments to the Author

Reviewer #1: (No Response)

7. PLOS authors have the option to publish the peer review history of their article (what does this mean?). If published, this will include your full peer review and any attached files.

Reviewer #1: No

---

## [Editor Report · Acceptance letter]

14 Jan 2021

PONE-D-20-03344R2 

Safety of antidepressants in a primary care cohort of adults with obesity and depression 

Dear Dr. Morriss:

I'm pleased to inform you that your manuscript has been deemed suitable for publication in PLOS ONE. Congratulations! Your manuscript is now with our production department. 

Kind regards, 

on behalf of

Dr. Nienke van Rein 

Academic Editor

PLOS ONE